# Teachers’ Health: How General, Mental and Functional Health Indicators Compare to Other Employees? A Large French Population-Based Study

**DOI:** 10.3390/ijerph191811724

**Published:** 2022-09-17

**Authors:** Mélèa Saïd, Sofia Temam, Stephanie Alexander, Nathalie Billaudeau, Marie Zins, Sofiane Kab, Marie-Noël Vercambre

**Affiliations:** 1MGEN Foundation for Public Health, 3 Square Max-Hymans, CEDEX 15, 75748 Paris, France; 2INSERM UMS 011, 94807 Villejuif, France; 3Faculty of Medicine, Université Paris Descartes, 75006 Paris, France

**Keywords:** teachers, employees, mental health, functional health, MSD, occupational health

## Abstract

Teachers’ health is a key factor of any successful education system, but available data are conflicting. To evaluate to what extent teachers’ health could be at risk, we used pre-pandemic data from the CONSTANCES population-based French cohort (inclusion phase: 2012–2019) and compared teachers (n = 12,839) included in the cohort with a random subsample selected among all other employees (n = 32,837) on four self-reported health indicators: perceived general health, depressive symptoms (CES-D scale), functional limitations in the last six months, and persistent neck/back troubles (Nordic questionnaire). We further restricted our comparison group to the State employees (n = 3583), who share more occupational similarities with teachers. Lastly, we focused on teachers and evaluated how their health status might differ across teaching levels (primary, secondary, and higher education). As compared to non-teacher employees, and even after adjusting for important demographic, socioeconomic, lifestyle, and occupational confounders, teachers were less likely to report bad perceived health and depressive symptoms but were more likely to present functional limitations. Trends were similar in the analyses restricted to State employees. Within the teaching population, secondary school teachers were more likely to report depressive symptoms but less frequently declared persistent neck/back troubles than primary school teachers. Our descriptive cross-sectional study based on a probability sampling procedure (secondary use of CONSTANCES inclusion data) did not support the idea that teachers’ health in France was particularly at risk in the pre-pandemic period. Both cross-cultural and longitudinal studies are needed to further gain information on the topic of teachers’ health around the world and to monitor its evolution over time, particularly during crises impacting the education system such as the COVID-19 pandemic.

## 1. Introduction

Teachers represent a large population of workers [1]. In France, around one million teachers invest themselves day after day in the training and education of future generations [2]. Their skills, but also more broadly their health and motivation [3], are key factors of any efficient education system [3,4]. Indeed, good health has been closely related to job satisfaction and motivation, low absenteeism, and high quality of work [5,6,7]. Furthermore, teacher wellbeing has been shown to lead to better student wellbeing and lower psychological difficulties through improved relationships with students in school and reduced teachers’ absenteeism [8]. Indeed, teachers, and schools more broadly, play a critical role in the social, cultural, moral, and health education of children, imparting the key values and competencies of a given society to prepare children to be active and healthy adults [9,10].

Yet, as a profession in contact with the public (students and their families), teachers are also exposed to specific occupational risks factors, mainly psychosocial [11]. Given this context, it is important to evaluate the different dimensions of teachers’ health to identify health promotion opportunities.

In the pre-pandemic literature on teachers’ health [12], certain topics were particularly recurrent and covered varied health dimensions: general health, mental health (depressive symptoms, burnout, stress, and anxiety), or functional health (“Ear, Nose, Throat” (ENT) disorders and musculoskeletal disorder (MSD)). Regarding general health, and although contexts and methods varied, studies suggested that, overall, teachers’ health was good compared to other professions [12,13,14], in line with the more holistic approach considering teacher wellbeing [15]. In France, an analysis of mortality (a very rough but informative indicator of health) by occupation and activity sector corroborated these findings, showing that teachers had a lower risk of mortality than the rest of the working population [16]. Nonetheless, some epidemiologic studies have highlighted certain mental and functional health problems to which this population could be more vulnerable [14,17,18,19,20], though evidence concerning mental health remains conflicting. For example, a French study of psychiatric disorders, assessed through the Composite International Diagnostic Interview–Short Form (CIDI-SF), concluded that teachers did not have poorer mental health compared to other employees [21], while a Flemish study based on the Short-Form 36 (SF-36) health survey tended to attest otherwise [22]. In addition, as human service professionals, teachers have also been found to be prone to burnout [23]. However, the many studies investigating this topic among teachers have had mixed results [24,25]. More than a real country-related disparity in the situation for teachers, differences in the comparison groups (if any) and in the instruments used to evaluate mental health may underlie such discrepancies.

Regarding functional health, alongside various health problems that can affect it in the general population (organ deficiency, mental health problems, etc.), teachers’ occupational exposures are closely associated with the risk of ENT disorders such as dysphonia, an alteration of the acoustic qualities of the voice. The voice is an indispensable tool for teachers, and ENT disorders, beyond having negative consequences for daily activities, may deteriorate the practice of their profession [26,27,28]. Finally, regarding MSD, a high proportion of teachers complain about musculoskeletal symptoms that affect their functional health and quality of life [29,30]. Interestingly, epidemiologic studies suggest that depression and psychosocial factors could be independently associated with both ENT [31] and MSD [32,33].

Although plethoric, available data on teachers’ health are of variable quality and any inconsistency in the reported results may be due to methodological differences, particularly regarding the selection of the comparison group and confounders, as well as to cultural differences. To evaluate to what extent teachers’ health could be at risk in a “normal” (pre-pandemic) time, and thus contribute to identifying opportunities for interventions, we drew on a large French nationwide probability sample of all-sector employees, among which we were able to situate teachers on four health indicators corresponding to general, mental, functional, and musculoskeletal health. Given previous French literature on teachers’ health [13,21] and teachers’ health behaviors [34,35], we hypothesized that: 1/ teachers’ health indicators will be better compared to non-teaching employees, even when adjusting for important confounders; 2/ within the teaching profession, teachers’ health will be homogeneous.

## 2. Materials and Methods

### 2.1. Study Population

The present cross-sectional study is part of the ESTER study (Étude Santé-Travail dans l’Enseignement et la Recherche; “Occupational health study in education and research”), which has been specifically designed within the CONSTANCES [36] cohort to investigate the occupational risks and work-related health determinants of education and research professionals, particularly teachers. The CONSTANCES cohort is a large French population-based prospective study of more than 210,000 participants. Adults aged 18 to 69 at inception were randomly selected from the National Health Insurance Fund (CNAMTS: Caisse Nationale d’Assurance Maladie des Travailleurs Salariés), which covers salaried workers (“employees”), professionally active or retired, and their families. It, thus, reaches more than 80% of the French population of this age range, only excluding self-employed or agricultural workers (entrepreneurs, business owners, and agricultural occupations) and undocumented immigrants. The cohort was approved by the Institutional Review Board of the National Institute for Medical Research—INSERM (no. 01–011) and has obtained the authorization of the National Data Protection Authority (Commission Nationale de l’Informatique et des Libertés (CNIL, no. 910486)). During the inclusion phase (2012–2019), eligible individuals were selected from the national social security database, which includes all employees living in France, according to a random sampling scheme stratified by age, gender, socio-economic status (SES), and region of France. These persons received an invitation sent to their homes explaining the aim of the CONSTANCES project and how data privacy and security were ensured. There were no financial reward/incentives nor compensation for participation: participation was entirely voluntary. Those who volunteered (7.3% of those invited) provided informed consent and underwent an extensive medical examination in a health screening center. They also completed various questionnaires relating to their personal, environmental, behavioral, occupational, social, and medical factors. The participation (enrollment) rate was of the same order of magnitude as other similar cohorts, such as for the UK Biobank [36].

As part of the ESTER project, we identified all education or research professionals (n = 16,510) among participants who were actively employed and enrolled in CONSTANCES until June 2019, and added to this exhaustive sample, a random sample twice as large of employees involved in sectors other than education and research (n = 33,020). Four participants withdrew their consent between the selection of the population (June 2019) and the data extraction (July 2019) and were, therefore, excluded from the ESTER population (Figure 1).

In the present study, we first focused on teachers’ health compared to that of other employees, and second, we examined their health according to teaching level. We excluded participants who did not fill out the lifestyle questionnaire (n = 280), as this was where our primary outcomes were reported. We also excluded research and non-teaching school staff from the group of education and research professionals (n = 3317) to focus exclusively on teachers. Finally, we excluded teachers whose teaching level was unknown (n = 253).

### 2.2. Health Indicators

For the sake of simplicity and homogeneity, we investigated four complementary binary indicators of self-reported health. Each of these binary indicators related to health dimensions recurrent in our literature review on teachers’ health and for which relevant information was available in the subjacent CONSTANCES primary setting, namely: general, mental, functional, and musculoskeletal health.

Among these four indicators, the two regarding general health and functional health were based on a single question (with a graded response), whereas the two other indicators (mental health/musculoskeletal health) were based on a validated questionnaire, each including several items.

To define the binary indicator of perceived general health, we considered the 8-point Likert scale “How would you describe your general health?” (quoted from “A-very good” to “H-very poor”) and used the central cut-off to broadly distinguish between “good perceived health” (A-D) vs. “bad perceived health” (E-H).

The mental health indicator was defined using the French version of the Center for Epidemiologic Studies-Depression (CES-D) scale [37]. This psychometric questionnaire of 20 items (score range: 0–60) allowed us to evaluate the extent of the depressive symptoms. In the present data, the Cronbach’s alpha of the CES-D was 0.897, confirming its good internal consistency. Based on the French validated sex-specific cut-off to identify persons at high risk of clinical depression [38], individuals with a score above 17 for men and 23 for women were classified as having depressive symptoms (“depressive symptoms” vs. “low to moderate”).

Regarding the functional health indicator, we focused on the limitations in daily activities as a positive response to the question: “Over at least the past 6 months, have you been limited, i.e., do you experience difficulties due to a health-related problem, in performing routine activities (at home, at work, during leisure activities, etc.) by comparison to other people of your age?”. This question had four response modalities—“yes, significantly limited”, “yes, limited”, “yes, slightly limited”, and “no”. In relation to the distribution of the variable, we defined as functionally limited those who declared being “significantly limited” or “limited” (“functional limitations” versus “none to slight”).

For the MSD indicator, and as school teachers appear to be particularly prone to suffer MSD of the upper limbs [17], we used the French version [39] of the standardized Nordic questionnaire [40] and focused on pain reported in the neck and/or the lower back with the intensity of the problem being ≥3 on a scale of 0 (no discomfort or pain) to 10 (maximum conceivable pain) that has lasted more than 30 days over a year, hereafter referred to as “persistent neck/back troubles”.

### 2.3. Covariates

In this comparison study of teachers and non-teachers, we considered various sociodemographic, socioeconomic, lifestyle, environmental, and occupational factors as potential confounders of the association between health and occupation. These health risk factors have been previously highlighted in France to be different among teachers as compared to the general population [34,35], and were available in the CONSTANCES inclusion dataset. All these covariates were recorded at enrollment through the lifestyle questionnaire or the medical examination. In fact, we aimed at capturing the differences in health among teachers and other non-teacher employees “all other things being equal” (sex, age category, education level, and so on).

For the sociodemographic factors, alongside gender and age categories (<35, 35–49, and ≥50 years old), we considered relationship status based on two questions: “Do you live with your partner?” and “Are you in a romantic relationship?”. The second question was considered if the answer to the first one was negative, to create the covariate “In a relationship, yes/no”. 

As socioeconomic factors, we considered education level and perceived financial difficulties. Education level was collected according to the International Standard Classification of Education and was then categorized into three levels: “high school diploma or less” (≤13 years of education), “undergraduate degree” (14–16 years of education), and “postgraduate degree” (≥17 years of education). Perceived financial difficulties were evaluated with the question “Are there times when you experience difficulties in meeting your financial needs (food, rent, electricity, loans, etc.)?”. Four answers were possible: “no, and it has never occurred”, “no, but it has happened in the past”, “yes, for less than a year”, and “yes, for several years”.

For the lifestyle indicators, we considered body mass index (BMI), smoking status, and alcohol consumption. Based on direct measures, we calculated BMI as weight divided by the square of the height in kg/m^2^ and categorized it into three modalities [41]: “underweight” (<18.5), “normal” (18.5–25), and “overweight/obesity” (>25). Smoking status was coded as “current smoker”, “former smoker”, and “non-smoker”. Average daily alcohol consumption was estimated using a self-reported table of average consumption per day of the week and per type of alcoholic beverage and was categorized in three modalities referring to the French national recommended limit at the time of data collection: “non-consumer”, “moderate consumption”, and “higher consumption than recommended” [42].

For the environmental factor, and to broadly characterize the living environment of the study population, we considered the degree of urbanicity of the commune of residence defined by the French National Institute of Statistics and Economics in four categories: “city center”, “suburb”, “isolated town”, and “rural municipality” (https://www.insee.fr/en/metadonnees/definition/c1902 (accessed on 1 August 2022)).

Finally, for the occupational factor, as a general psychosocial risk measure at work available in the CONSTANCES inclusion data, we considered the effort–reward ratio, derived from the effort–reward imbalance at work model (ERI), which is a “psychometrically well-justified measure of work-related stress grounded in sociological theory” [43]. In the ERI model, chronic work-related stress is identified as “non-reciprocity or imbalance between high efforts spent and low rewards received”. Rewards may be provided in terms of money, esteem, and career opportunities (including job security). The ratio was dichotomized into “≤1” (i.e., lower or equal effort than reward = ‘not at risk’) and “>1” (i.e., higher effort than reward = ‘at risk’).

### 2.4. Missing Values

In our study, rates of missing values for the four health indicators ranged from 1.2% (functional limitations) to 5.0% (depressive symptoms). They were slightly but systematically lower among teachers than among other employees. For each of these outcomes, participants with a missing value were alternately excluded from the corresponding model, resulting in a slightly different sample size each time. 

Regarding the covariates, there were no missing values for age, gender, and degree of urbanicity. For the other covariates, when the rate of missing values was <5% (i.e., relationship status, perceived financial difficulties, smoking status, and education level), in order to limit sample attrition, we imputed them by the most common value observed in the entire sample or, specifically for the education level, according to the gender, age category, and according to whether they were teachers or not. Otherwise, we created a “missing” category (i.e., for BMI, alcohol consumption, and effort–reward ratio). 

### 2.5. Statistical Analyses

To begin, we compared the characteristics and the indicators of ill-health for teachers (n = 12,839) to the whole “non-teacher employees” group (G1, n = 32,837) and then to the restricted group of “non-teacher State employees” (G2, n = 3583). As teachers were all State employees in France, this later comparison group was identified to refine the analysis, by limiting de facto socioeconomic and occupational heterogeneity. Then, within the teachers’ sample, to explore the extent to which teachers’ health was or was not homogeneous across grade levels taught, we compared the health indicators according to teaching level: primary school (n = 4877 teachers), secondary school (n = 6466 teachers), and higher education (i.e., university) (n = 1496 teachers).

We used a chi^2^ test to compare the prevalence of the four health indicators between teachers and the two comparison groups alternately, and across teachers’ grade levels. For each health indicator of interest, we implemented two logistic regression models. The first model was adjusted for gender and age (M1). The second model was further adjusted for all the other covariates described above (M2). 

We conducted several sensitivity analyses to assess the robustness of our results. First, we explored to what extent the associations observed in the main analysis (pooled data from several years of CONSTANCES inclusion) may evolve over time, in implementing a stratification analysis by years of data collection: 2012/2013/2014; 2015/2016; 2017/2018. Second, to evaluate to what extent our imputation choices could have influenced the results, we implemented the models described above: 1/ without imputing missing values for the covariates (“complete case analysis”); 2/ using a multiple imputation procedure for covariates. We compared the coefficients thus obtained to the models involving ad hoc imputation of covariates. Third, we used a more stringent definition criterion for the three health indicators studied that used an ad hoc cutoff. That is, we considered the indicator of “very bad health” if the person quoted F, G, or H to the question “How would you rate your general health?”; we considered only the response “strongly limited” to define “strong functional limitations”; we considered only those quoting the musculoskeletal pain level of ≥4 (instead of 3) to define “severe persistent neck/back troubles”. We then conducted a stratified analysis in the fully adjusted model among men (n = 4285 male teachers vs. n = 15,952 other male employees) and women (n = 8554 female teachers vs. n = 16,885 other female employees). Analyses were run using STATA version 16 (StataCorp. 2019. Stata Statistical Software: Release 16. College Station, TX, USA: StataCorp LLC.).

## 3. Results

### 3.1. Characteristics of Teachers Compared to Non-Teacher Employees

We observed some specificities of the teacher group compared to the two non-teacher comparison groups, namely “non-teacher employees” (G1) and “non-teacher State employees” (G2) (Table 1).

Teachers were largely female (66.6% among teachers vs. 51.4% among G1 and 60.6% among G2). Their socioeconomic background appeared consistently more advantaged than the other groups: they were more likely to have a high level of education (43.8% were postgraduate among teachers vs. 25.5% among G1 and 25.8% among G2) and less likely to report financial difficulties (73.7% of teachers declared never having financial difficulties vs. 62.2% among G1 and 63.9% among G2). Furthermore, teachers’ lifestyle indicators were healthier with a higher likelihood of having a normal BMI (65.7% among teachers vs. 56.5% among G1 and 57.6% among G2), of being non-smokers (56.3% among teachers vs. 46.8% among G1 and 50.2% among G2), and of consuming alcohol moderately (77.1% among teachers vs. 74.8% among G1 and 74.6% among G2). Regarding psychosocial risk at work, we did not observe a major difference in the effort–reward imbalance across occupational groups under study.

### 3.2. Health Indicators of Teachers Compared to Non-Teacher Employees

Regarding the four complementary outcomes related to general, mental, functional, and musculoskeletal health, prevalence of ill-health was systematically lower among teachers than among the two other employee groups (Table 2). 

Models adjusted for age and gender confirmed these raw observations, with teachers less likely than non-teacher employees to declare being in bad health, to have depressive symptoms, to be functionally limited, or to report persistent neck/back troubles. 

In the fully adjusted models, associations were weaker, and some were no longer significant. In particular, there were no longer significant differences in persistent neck/back troubles between teachers and both non-teacher employees groups. Still, as compared to non-teacher employees, teachers were less likely to declare being in bad health (adjusted odds ratio [95% confidence interval]: OR [95%CI] = 0.81 [0.73;0.89]; when the comparison group was restricted to State employees, OR [95%CI] = 0.83 [0.69;0.99]). Teachers were also less likely to have depressive symptoms than the other employees (OR [95%CI] = 0.92 [0.85;1.00]) and the State employees (OR [95%CI] = 0.84 [0.73;0.97]). However, teachers were more likely to have functional limitations than other non-teacher employees (OR [95%CI] = 1.12 [1.04;1.21]), this difference being no longer significant when the comparison group was restricted to State employees. In further analyses among employees having reported functional limitations, we explored potential differences between teachers and non-teachers in terms of reasons invoked. When classifying the reasons into seven main categories (accident, mental health, visual or hearing impairment, chronic disease, significant pain, headache, excess weight, and other), and after adjustment for potential confounders, no one category of reasons seemed to predominantly account for the greater likelihood of teachers to report a limitation: whereas excess weight was clearly underrepresented among reasons invoked by teachers (OR [95%CI] = 0.57, *p* < 0.001), the other categories were slightly more frequently invoked by teachers than non-teachers with the same demographic, socioeconomic, and lifestyle profile, but no association reached significance level.

### 3.3. Comparison of Health Indicators among Teachers according to Teaching Level

In the analysis restricted to teachers, we observed a certain homogeneity across the teaching levels of the four health indicators studied (Table 3), with two exceptions. 

Compared to primary school teachers, secondary school teachers were more likely to have depressive symptoms when taking into account potential confounders (OR [95%CI] = 1.35 [1.16;1.57]). On the other hand, they were slightly less likely to have persistent neck/back troubles compared to primary school teachers (OR [95%CI] = 0.91 [0.83;1.00]). 

### 3.4. Sensitivity Analyses

In the stratification analysis according to years of data collection (subsample 1, data collected in 2012/2013/2014, and N = 3096 teachers vs. 9291 non-teachers; subsample 2, data collected in 2015/2016, and N = 4744 teachers vs. 11,355 non-teachers; subsample 3, data collected in 2017/2018, and N = 4916 teachers vs. 12,218 non-teachers, respectively), results were quite stable across the three subsamples, although we noticed an attenuation of the difference in mental health (to the disadvantage of teachers).

In the sensitivity analyses implemented in the sample without any missing values, or alternately, using multiple imputation, differences between odds ratios or interval boundaries were small and trends were unchanged.

When using more stringent criteria to define the ill-health indicators, results were consistent with the main analyses, although certain associations were no longer significant, specifically for the outcomes “very bad health” and “severe persistent neck/back troubles” (results not shown in tables).

In the analysis by gender, we occasionally observed some evidence of an interaction effect, where differences across occupational groups under study, if any, were significant among women only (Table 4).

## 4. Discussion

In this large population-based study of French employees conducted before the COVID-19 pandemic, even after adjusting for various potential confounders, teachers were less likely to report being in bad health and having depressive symptoms than non-teacher employees, supporting the hypothesis that the normal (pre-pandemic) health status of teachers in France was not particularly at risk. Among teachers, general and functional health indicators were similar across teaching levels, validating the pragmatic approach of considering teachers a single entity in the first step of comparison. Only secondary school teachers were more prone to depressive symptoms, but they were less likely to report persistent neck/back troubles than primary teachers.

### 4.1. What Lessons Are Learned about Pre-Pandemic Teachers’ Health Status in France?

Overall, our results of a better general and mental health status among teachers when compared to other employees are consistent with our hypothesis and with the few previous scientific studies conducted in France in the pre-pandemic period [11,13,21]. Different hypotheses can be put forward to explain these findings. First, the better health status of teachers could be linked to a selection effect at profession entry in which healthy people will be particularly favored. Indeed, persons who become teachers have studied several years after graduating from high school, passed selective exams, and are generally aware of health and education issues. Such self-selection and trajectories are associated with a higher socioeconomic background, greater health literacy, and healthier behaviors [34,35], all factors linked with better health. Second, healthier living and working environments as well as healthier lifestyles throughout their careers would maintain or increase the initial teachers’ health potential [44]. Despite established occupational risk factors such as violence in schools [45] or a feeling of a lack of social recognition among French teachers [46], some positive factors such as the social support received from the teaching team [47], dispositional happiness [48], or the reward of seeing one’s students’ progress could decisively contribute to good perceived health [49]. Taking into account certain specific confounding factors such as education level or punctual health behaviors would not be sufficient to correct for such a selection effect. It would be interesting to determine in a long-term longitudinal setting which of the two phenomena—selection on profession entry or preservation/improvement during the career—is better able to explain the teachers’ health distinction in France.

With regard to certain persistent alarmist claims in France about teachers’ health, particularly concerning mental health dimensions (e.g., “teachers are more depressed than average”), these could be explained by the fact that teachers are particularly “exposed” to the public through their students: when a teacher is absent, several families, in addition to the institution, are impacted, whereas for the great majority of office employees, absenteeism is less visible [28]. Noteworthy is the result that teachers were more likely to present functional limitations than non-teacher employees with the same demographic, socioeconomic, and lifestyle profile. Among the different categories of reasons invoked for functional limitations (vision or hearing impairment, mental health, chronic disease, etc.), this greater likelihood of teachers to be functionally limited as compared to non-teachers could not be attributed to a specific background. Interestingly, the trend was reversed in the age- and sex-adjusted model and no longer significant when the comparison was restricted to State employees. Rather than a higher risk of disability due to working conditions in schools (i.e., “work hazard”), we hypothesized that this statistical association can be explained by an interaction between self-selection and a better inclusion of persons with a disability in education than in other (private) sectors (in terms of accessibility at profession entry and/or accompaniment throughout the career). An analysis of how the events “limitation onset” and “entry into the profession” are sequenced over time would be very insightful in this regard.

When exploring to what extent the health differences between teachers and non-teachers may evolve over the 6 years of data collection, results were quite stable, justifying a posteriori the pooled approach. We nonetheless noticed a slight attenuation of the difference in mental health to the disadvantage of teachers that calls for an in-depth study based on longer-term longitudinal data. Preliminary data have pointed to a slight but continual decline in the mental health of French teachers well before the COVID-19 pandemic [50].

We observed an isolated interaction effect between gender and teaching occupation regarding depressive symptoms, where the teachers’ lower risk of depressive symptoms was significant among women only. We noted the same interaction with perceived health, but it was less marked. Consistent with a previous study of human service professions [51], these observations point to gender-related differences in selection/evolution mechanisms during the teaching career [52], which remain to be clarified.

### 4.2. What Lessons Are Learned about the Homogeneity of Teachers’ Health by Teaching Level?

Although teachers’ mental health appears good compared to other employees with the same demographic and socioeconomic background in the pre-pandemic time, we observed that secondary school teachers were more likely to report depressive symptoms than primary school teachers. This is consistent with a previous comparison in France of psychiatric disorder prevalence among teachers [53], as well as previous studies showing higher exposure to depression risk factors in secondary schools, in particular, school violence and bullying [45]. Regarding musculoskeletal health, whereas differences between teachers and non-teacher employees were not significant, we observed that primary school teachers were more likely to report persistent neck/back troubles than secondary school teachers, as already suggested in the literature [19]. This could be related to the greater physical demands when caring for younger (smaller) children. In order to explore this hypothesis, it would be interesting to refine the gradient of teaching levels, for instance, to compare MSD complaints of kindergarten vs. elementary school teachers; however, we did not have this level of precision in our data. Our observations require further study to better target prevention strategies at the higher-risk teaching occupations.

### 4.3. Limitations and Perspectives

Our results should be read keeping in mind the limitations and strengths of this study, based on the secondary use of generalist pre-pandemic data. The main strengths are those of the underlying CONSTANCES dataset, a large population-based cohort intended to serve as an epidemiological research infrastructure accessible to the epidemiologic research community, in particular, in the field of occupational health [36]. Its strengths include: a large sample size, the availability of complementary high-quality data on various health dimensions, work characteristics, and potential confounders in the association network between health and occupation. Nonetheless, in the context of secondary data use, some specific determinants or outcomes particularly relevant to the education sector (for example, the precise teaching level, the subject taught by the teacher, and data on dysphonia) were missing, and the choice of outcomes was somewhat constrained. Another limitation relates to participation bias in the CONSTANCES cohort. Indeed, the data analyzed here correspond to the enrollment step of an epidemiologic cohort. As active and healthy persons are more prone to participate in studies that require a certain amount of commitment [54], prevalence of diseases is often underestimated at inclusion (i.e., “healthy participant bias”). However, our objective was to test multi-adjusted statistical associations (i.e., between health indicators and teaching occupation), not to estimate/extrapolate raw prevalence rates. Such an association analysis is less sensitive to participation bias. Indeed, it is unlikely that individuals who did not respond, because they were in poor health, would be proportionately more present in one occupation group than in the other.

Another point for discussion is the constitution of the comparison group. Here, we assessed the differences between teachers, a relatively homogeneous socio-professional group to a much more socioeconomically varied group of employees working in other sectors than education and research. To better circumscribe the specificities of teachers’ health and given that they are all employed by the State, we chose to refine the comparison in a second step by restricting the analysis to those employed by the State, thus reducing the variability linked to employment. Results corroborated the finding that the teachers’ health particularities do not entirely result from other parameters. Regarding mental health, we even observed a greater distinction between teachers and other State employees than between teachers and all other employees, suggesting that the mental health of non-teaching State employees is poorer than average, a finding that requires further study.

In addition, health status cannot be entirely apprehended in its complexity by four, albeit complementary, indicators, all based on self-report. Other dimensions (e.g., gastrointestinal illness [55]) or indicators (e.g., anxiety [56]), which could be objectively evaluated, would be important to consider to better inform the body of knowledge about teacher health. Nonetheless, the relevance of considering subjective health has been amply documented and its study remains not only convenient but also informative, in particular, in the field of mental health and general well-being [57].

Finally, our results are based on French pre-pandemic data and generalizability is not a matter of course. Given the heterogeneity in teachers’ sociodemographic and cultural backgrounds across countries [58,59], cross-cultural studies are needed to expand the view on teacher health around the world. Furthermore, the present analysis applies to a time before the COVID-19 pandemic. How this major crisis, throughout which teachers played a key role, will impact the pre-pandemic health status of populations and the wellbeing differences between occupations will need to be thoroughly investigated and even more so as recent preliminary studies in various countries have pointed to a concerning evolution of teachers’ mental health throughout the pandemic [60,61,62]. In that context, our study provides a clear first view of teachers’ health in France in pre-pandemic times, and it also opens important reflections that are relevant globally, namely about the importance of properly assessing teachers’ health and monitoring its evolution over time to identify opportunities for interventions.

## 5. Conclusions

In a large nationwide sample of French employees, teachers reported better general and mental health than non-teachers with a similar demographic, socioeconomic, lifestyle, and environment profile. Given the major role of teachers for society, both cross-cultural and longitudinal studies are needed to inform the topic of teacher health around the world, to monitor its evolution over time, particularly during crises impacting the education system, and to better understand the underlying mechanisms.

## Figures and Tables

**Figure 1 ijerph-19-11724-f001:**
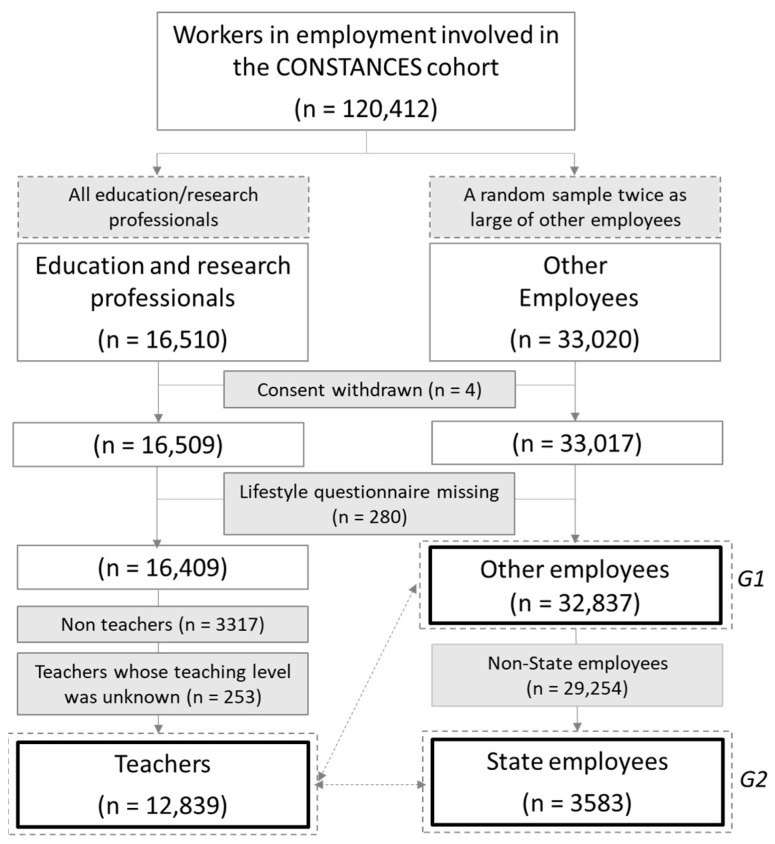
Flowchart of the study population.

**Table 1 ijerph-19-11724-t001:** Characteristics of teachers compared to all non-teacher employees and to other State employees, ESTER-CONSTANCES study.

		Teachers % (N = 12,839)	Non-Teacher Employees	
			All (G1) % (N = 323,837)	State Employees (G2 ^1^) % (N = 3569)
Gender	Men	33.4	48.6 ***	39.4 ***
	Women	66.6	51.4	60.6
Age	<35	14.1	25.8 ***	15.8 ***
	35–49	55.2	42.9	42.1
	≥50	30.7	31.3	42.1
Relationship status	In a relationship	80.3	74.4 ***	71.3 ***
	Single	19.7	25.6	28.7
Education level	High School diploma or less	1.4	39.0 ***	34.3 ***
	Undergraduate degree	54.7	35.6	39.9
	Postgraduate degree	43.8	25.5	25.8
Perceived financial difficulties	Never	73.7	62.2 ***	63.9 ***
	Past only	19.9	26.1	25.8
	This year	3.2	6.0	4.7
	Several years	3.3	5.8	5.6
BMI	Underweight	3.6	2.4 ***	2.4 ***
	Normal BMI	60.5	51.9	54.3
	Overweight/Obesity	27.9	37.7	37.6
	Missing	8.0	8.0	5.8
Smoking status	Non-smoker	56.3	46.8 ***	50.2 ***
	Current Smoker	13.3	20.9	17.0
	Former smoker	30.4	32.3	32.8
Alcohol consumption	None	13.9	13.7 ***	15.3 *
	Moderate	70.6	67.0	66.8
	At risk	7.0	8.8	7.4
	Missing	8.5	10.5	10.5
Degree of urbanicity	Suburbs	33.2	30.9 ***	31.7 ***
	City center	42.4	43.8	48.2
	Isolated town	6.5	6.9	6.4
	Rural municipality	17.9	18.4	13.7
Effort–reward ratio ^2^	≤1	50.7	47.7 *	48.9 NS
	>1	46.6	46.3	48.2
	Missing	2.7	6.0	2.9

^1^ Group G2 is a subpopulation of G1. ^2^ From the effort–reward imbalance questionnaire (Siegrist et al. 2004 [43]). *, *** Prevalence significantly different from teachers’ at 5% level, and 0.1% level, respectively; NS = not significant; for other variables than BMI, alcohol consumption, and effort-reward ratio, rate of missing value < 5%, and missing values were not considered in the distribution.

**Table 2 ijerph-19-11724-t002:** Health indicators of teachers compared to all non-teacher employees and to other State employees, basic and further adjustment, ESTER-CONSTANCES study.

	Teachers	Non-Teacher Employees			Teachers vs. G1	Teachers vs. G2 ^1^
		**All (G1)**	**State Employees (G2 ^1^)**			
**Health indicator**	**%**	**%**	**%**		**OR [95%CI]**	**OR [95%CI]**
**Bad perceived health**	5.1	9.1 ***	9.1 ***	M1	0.50 *** [0.45;0.54]	0.55 *** [0.47;0.63]
(*vs. good*)				M2	0.81 *** [0.73;0.89]	0.83 * [0.69;0.99]
**Depressive symptoms**	8.8	12.9 ***	13.6 ***	M1	0.66 *** [0.61;0.71]	0.63 *** [0.56;0.71]
*(vs. low to moderate)*				M2	0.92 * [0.85;1.00]	0.84 * [0.73;0.97]
**Functional limitations**	10.2	11.7 ***	11.7 **	M1	0.79 *** [0.73;0.84]	0.87 * [0.77;0.98]
*(vs. none to slight)*				M2	1.12 ** [1.04;1.21]	1.09 [0.94;1.26]
**Persistent neck/back troubles**	23.2	26.8 ***	26.5 ***	M1	0.75 *** [0.71;0.79]	0.83 *** [0.76;0.91]
*(vs. no)*				M2	0.96 [0.91;1.02]	0.98 [0.88;1.09]

*, **, *** significantly different from non-teachers’ at 5% level, 1% level, and 0.1% level, respectively. Odds ratios and 95% confidence interval (OR [95%CI]) from logistic regressions. Model 1 (M1): adjusted for age and gender; Model 2 (M2): further adjusted for relationship status, education level, perceived financial difficulties, BMI, smoking status, alcohol consumption, degree of urbanicity, and effort–reward ratio. All the variable adjustments were categorized as documented in Table 1. ^1^ Group G2 is a subpopulation of G1.

**Table 3 ijerph-19-11724-t003:** Health indicators of teachers according to their teaching level, basic and further adjustment, ESTER-CONSTANCES study.

Health Indicator	%	M1—OR [95%CI]	M2—OR [95%CI]
**Bad perceived health** *(vs. good)*			
Primary school teachers	4.9	ref.	ref.
Secondary school teachers	5.5	1.08 [0.91;1.29]	1.11 [0.92;1.34]
Higher education teachers	3.9	0.79 [0.58;1.07]	0.82 [0.59;1.15]
**Depressive symptoms** *(vs. low to moderate)*			
Primary school teachers	7.4	ref.	ref.
Secondary school teachers	9.8 ***	1.25 ** [1.08;1.44]	1.35 *** [1.16;1.57]
Higher education teachers	9.0 *	1.09 [0.88;1.36]	1.12 [0.87;1.43]
**Functional limitations** *(vs. none to slight)*			
Primary school teachers	10.3	ref.	ref.
Secondary school teachers	10.7	1.04 [0.92;1.18]	1.10 [0.97;1.26]
Higher education teachers	7.1 ***	0.71 ** [0.57;0.89]	0.82 [0.64;1.04]
**Persistent neck/back troubles** *(vs. no)*			
Primary school teachers	26.1	ref.	ref.
Secondary school teachers	22.0 ***	0.85 *** [0.77;0.93]	0.91 * [0.83;1.00]
Higher education teachers	19.2 ***	0.78 ** [0.67;0.91]	0.90 [0.76;1.06]

*, **, *** significantly different from primary school teachers at 5% level, 1% level, and 0.1% level, respectively. Odds ratios and 95% confidence interval (OR [95%CI]) from logistic regressions. Model 1 (M1): adjusted for age and gender; Model 2 (M2): further adjusted for relationship status, education level, perceived financial difficulties, BMI, smoking status, alcohol consumption, degree of urbanicity, and effort–reward ratio. All the variable adjustments were categorized as documented in Table 1.

**Table 4 ijerph-19-11724-t004:** Gender stratification and interaction between gender and teaching occupation on health indicators, ESTER-CONSTANCES study.

	Teachers vs. Non-Teacher Employees (G1)		
Health Indicator	Among Men—OR [95%CI]	Among Women—OR [9 5%CI]	*p*-Interaction
**Bad perceived health** *(vs. good)*	0.89 [0.75;1.05]	0.77 *** [0.68;0.87]	0.09
**Depressive symptoms** *(vs. low to moderate risk)*	1.04 [0.92;1.18]	0.86 ** [0.77;0.96]	0.00
**Functional limitations** *(vs. none to slight)*	1.15 * [1.01;1.32]	1.11 * [1.01;1.22]	0.95
**Persistent neck/back troubles** *(vs. none)*	0.98 [0.89;1.08]	0.96 [0.90;1.02]	0.92

*, **, *** significantly different from non-teachers’ at 5% level, 1% level, and 0.1% level, respectively. Odds ratios and 95% confidence interval (OR [95%CI]) from logistic regressions. Adjusted for age, gender, relationship status, education level, perceived financial difficulties, BMI, smoking status, alcohol consumption, degree of urbanicity, and effort–reward ratio. All the variable adjustments were categorized as documented in Table 1.

## Data Availability

The data underlying this article were provided by the CONSTANCES cohort under permission and are not publicly available, to ensure the confidentiality of study participants. However, the CONSTANCES cohort is “an open epidemiological laboratory” and access to study protocols and data is available on request (www.constances.fr/conduct-project-ongoing.php (accessed on 1 August 2022)).

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
