# Peer review of "Teachers’ Health: How General, Mental and Functional Health Indicators Compare to Other Employees? A Large French Population-Based Study"

_ijerph, 2022, doi:10.3390/ijerph191811724_

Round 1

Reviewer 1 Report

Thank you for the opportunity to review this work. The authors are commended for addressing such a timely issue and expanding the conversation regarding teachers’ health – particularly in encompassing the varying perspectives of “health” (e.g., physical, mental, functional, occupational, etc.).

·      The authors do a nice job in setting up the rationale for the study and considering teachers’ physical and mental health.

·      The flow chart provides the reader with a clear perspective of how the sample of participants were extracted from such a large dataset and data collection effort.

·      When considering this data as a secondary dataset, what are the potential benefits and limitations of using this data?

·      Page 4, perceived general health indicator; How many points were on the Likert scale? As written is a little confusing by describing a binary indicator but describing points A-H.

·      What are the psychometrics of the measures used? Internal reliability (Cronbach’s alpha)? Are their subscales? Are these pre-established measures or researcher-created? Were some of these indicators just one question measures? What is the rationale behind using a one item measure? …. Perhaps it’s the word “indicator” that is confusing? Nonetheless, some greater clarity could be beneficial in this area.

·      How are participants recruited in this data collection? Is there a benefit or reward for participants engaging in this study?

·      What is the occupational factor measuring? What would be an example of psychosocial risk?

·      For the missing values, would the authors consider this “missing at random”? If using available case analysis, does this present or potentially prevent bias in the dataset? What is the rationale for the imputation methods were used for the missing values as mentioned on page 5-6?

Reviewer 2 Report

The article《Teachers’Health: How General, Mental and Functional Health Indicators Compared to Other Employees? A Large French Population-Based Study》has proposed four self-reported health indicators to explore and analyze the differences in physical and mental functional health between teachers and non-teachers in France before the pandemic, and discussed various possible moderating variables. Differences in health status between groups of teachers at different levels of teaching (primary, secondary and higher education)) were also compared. The topic selection is innovative to some extent, but the author needs to further modify the article because of the following problems.

Comment 1:Inappropriate use of data. In this study, the author selected the subjects for a total of 7 years from 2012 to 2019, but conducted a cross-sectional analysis. There may be interference from major events. And further accurate or longitudinal analysis of the subjects is needed.

Comment 2:The health indicators used are not comprehensive. The four health indicators used in this paper are all in the form of self-report. The subjective perception of mental health status may be inconsistent with the actual status due to individual sensitivity and other reasons. It is suggested to add the systematic information such as the recent physical examination report and medical records of the participants as a supplement to the assessment of mental health status.

Comment 3:The significance and practical value of the research need to be deeply explored. The research content of this study is the differences in physical and mental functional health between teachers and non-teachers in France before the pandemic, which is not enough in depth compared with more than 30,000 subjects. It is suggested that the author conduct an in-depth analysis of the reasons behind the differences and add supplements in the discussion section of 4.2.

Comment 4:The sample population was not comprehensive enough. It is suggested that the authors may supplement other countries and post-pandemic subjects to further enhance the universality and ecological validity of the results of this study.

Comment 5:The introduction of regulatory control variables requires further elaboration. The introduction of this paper only explains the teacher group and the four health indicators, without explaining the relevant effects of the moderating and controlling variables in the following model construction, such as gender and education background, etc. Further supplementary evidence is needed.

Comment 6:There are conflicting interpretations of the findings. In response to the results of this study, the authors suggest that the sensitivity of the teacher group may be a reason for their reporting more limitations, but cannot explain the sensitivity of the teacher group to mental illness. Therefore, it is recommended that the authors refine the interpretation of each study result and give comprehensive and systematic hypotheses and explanations at different levels.
